# Estimating All-Cause Deaths Averted in the First Two Years of the COVID-19 Vaccination Campaign in Italy

**DOI:** 10.3390/vaccines12040413

**Published:** 2024-04-13

**Authors:** Giovanni Corrao, Gloria Porcu, Alina Tratsevich, Danilo Cereda, Giovanni Pavesi, Guido Bertolaso, Matteo Franchi

**Affiliations:** 1National Centre for Healthcare Research and Pharmacoepidemiology, University of Milano-Bicocca, 20126 Milan, Italy; giovanni.corrao@unimib.it (G.C.); alina.tratsevich@unimib.it (A.T.); matteo.franchi@unimib.it (M.F.); 2Unit of Biostatistics, Epidemiology and Public Health, Department of Statistics and Quantitative Methods, University of Milano-Bicocca, 20126 Milan, Italy; 3Specialization School of Health Statistics and Biometrics, University of Padua, 35131 Padua, Italy; 4Preventive Unit of Welfare Department, Lombardy Region, 20124 Milan, Italy; danilo_cereda@regione.lombardia.it; 5General Directorate of Welfare Department, Lombardy Region, 20124 Milan, Italy; giovanni_pavesi@regione.lombardia.it; 6Welfare Department, Lombardy Region, 20124 Milan, Italy; guido_bertolaso@regione.lombardia.it

**Keywords:** COVID-19, vaccination, public health, epidemiology, excess all-cause deaths, SARIMAX

## Abstract

Comparing deaths averted by vaccination campaigns is a crucial public health endeavour. Excess all-cause deaths better reflect the impact of the pandemic than COVID-19 deaths. We used a seasonal autoregressive integrated moving average with exogenous factors model to regress daily all-cause deaths on annual trend, seasonality, and environmental temperature in three Italian regions (Lombardy, Marche and Sicily) from 2015 to 2019. The model was used to forecast excess deaths during the vaccinal period (December 2020–October 2022). We used the prevented fraction to estimate excess deaths observed during the vaccinal campaigns, those which would have occurred without vaccination, and those averted by the campaigns. At the end of the vaccinal period, the Lombardy region proceeded with a more intensive COVID-19 vaccination campaign than other regions (on average, 1.82 doses per resident, versus 1.67 and 1.56 in Marche and Sicily, respectively). A higher prevented fraction of all-cause deaths was consistently found in Lombardy (65% avoided deaths, as opposed to 60% and 58% in Marche and Sicily). Nevertheless, because of a lower excess mortality rate found in Lombardy compared to Marche and Sicily (12, 24 and 23 per 10,000 person-years, respectively), a lower rate of averted deaths was observed (22 avoided deaths per 10,000 person-years, versus 36 and 32 in Marche and Sicily). In Lombardy, early and full implementation of adult COVID-19 vaccination was associated with the largest reduction in all-cause deaths compared to Marche and Sicily.

## 1. Introduction

The clinical impact of a given medical intervention in a specific population (i.e., the fraction of clinical outcomes avoided as a result of the intervention in that population) is influenced by two quantities, namely effectiveness (i.e., the ratio between the outcome risk observed in those who received and those who did not receive the intervention) and prevalence in the specific population of individuals needing the intervention and actually receiving it [1]. In terms of the recent worldwide vaccination campaigns against COVID-19 (the medical intervention of interest), both quantities are known.

Regarding effectiveness, after COVID-19 vaccines achieved licensure, observational investigations carried out during ongoing vaccination campaigns consistently showed that mRNA-based and adenovirus-vectored vaccines are protective in the real world; this protection includes the risk of SARS-CoV-2 infection and re-infections, and, to an even greater degree, severe and lethal manifestations of the disease [2,3,4,5,6]. Observational studies also informed that the protective action declines over time [7,8], varies according with circulating variants [9,10,11,12,13], and varies according to whether one, two, or three doses were received [14,15]. Concerning prevalence, data on the progression of vaccination rates were made readily available from several worldwide organization databases [16,17,18], as well as from several country and regional open-access data repositories [19], thus allowing the measurement, monitoring, and comparison of the speed of progression of vaccination campaigns [20].

For several reasons, however, this impressive amount of data and evidence does not provide a complete picture of how many deaths have been avoided, both directly and indirectly, by vaccination campaigns. In fact, mortality rates attributed to COVID-19: (i) vary over time and across countries depending on the criteria for attributing deaths to the SARS-CoV-19 contagion [21,22,23] and testing availability [24]; (ii) lack death counts missed because they occurred outside of health facilities [25]; and (iii) exclude non-COVID deaths which are indirectly a result of the pandemic (e.g., because of disrupted health services) [26]. Therefore, several international organizations, including the World Health Organization (WHO) [27,28], suggest counting “all-cause excess deaths” to reliably capture the total impact of pandemics. For the same reasons, the fraction of all-cause excess deaths avoided by vaccination campaigns may capture the clinical impact of the campaign better than conventional measures. However, because other time-changing determinants of mortality are expected beyond exposure to the vaccination campaign (e.g., very cold environmental temperatures, heat waves [29], and the emergence of new variants that demonstrate vaccine escape [30]), these changes must be considered to model the burden of vaccine net of other factors.

In Italy, the vaccination campaign began following an order of priority based on the risk of infection and severe disease for COVID-19 in different population groups. The priority categories defined in the “Plan strategy for anti-SARS-CoV-2/COVID-19 vaccination” were as follows: health and social-health workers, residents and staff of residential facilities for the elderly, and people of advanced age (80+) [31]. Vaccination has been extended, with the increase in authorized vaccines and available doses, to other population groups, such as people with at least one chronic comorbidity, school personnel or law enforcement. Subsequently, vaccination was made available to all age groups in descending order.

The effectiveness of vaccines in the Italian setting was assessed by observational studies which showed a lowered risk of both SARS-CoV-2 infection and COVID-19-related outcomes in vaccinated as compared to unvaccinated people against both Alpha and Delta variants [11,32], as well as against the Omicron variant [33,34,35]. Moreover, during the year 2011, it was estimated by the National “Istituto Superiore di Sanità” that the vaccination campaign avoided a total of about 2.8 million cases of SARS-CoV-2, about 290,000 hospitalizations, about 37,700 hospitalizations in intensive care and about 77,700 deaths, which represented 43%, 58%, 57% and 64% of expected events in case of no vaccination, respectively [36].

Given the lack of evidence about the estimate of the impact of the Italian vaccination campaign in averting all-cause deaths directly and indirectly due to the SARS-CoV-2 virus, the aim of this study was to estimates the trend over time in the proportion of all-cause excess deaths avoided through the first two years of the vaccination campaign implemented in three regions of Italy. The regions differ in the speed with which candidates were reached with the first, second, and third doses of the vaccine, intensity and type (dominant variant) of SARS-CoV-2 spread, and other environmental factors.

## 2. Materials and Methods

### 2.1. Study Design and Population

An observational ecological study based on populations from three regions located in northern (Lombardy), central (Marche), and southern (Sicily) Italy was performed. Overall, the population of these regions included more than 13 million beneficiaries of the Italian National Health Service (NHS) aged 20 years or older, comprising 27% of the Italian population of that age category. A geographical map of the investigated regions, including age structure, is presented in Appendix A.

### 2.2. Data Sources

Official data on all-cause deaths occurring from 1 January 2015 to 31 October 2022 are made publicly available by the Italian National Institute of Statistics [37]. Official data on citizens who tested positive to PCR test to the SARS-CoV-2 virus in any clinical setting regardless of the presence of symptoms [38] from 27 February 2020 (the date of case zero identification in Italy), until 31 October 2022, and those who received one (first complete dose), two, and three doses (first and second booster doses, respectively) of the vaccine against COVID-19 from 27 December 2020 (vaccination campaign starting day in Italy) to 31 October 2022, are made publicly available by the Italian Health Ministry [18]. Data on daily environmental temperatures in the period 2015–2022 were gathered from the Italian National Environmental Protection System official site [39].

### 2.3. Estimating the Expected Excess All-Cause Deaths

Because daily death counts form a time process with “natural” variations on a seasonal basis, we used a seasonal autoregressive integrated moving average with exogenous factors (SARIMAX) model [40] to fit the data during the control period preceding the vaccination period (from 2015 to 2019, excluding the pandemic period uncovered by vaccination) separately for each region. Time series data on daily environmental temperatures, other than seasonality and annual trend, were considered in model fitting. The model was used to forecast the number of expected deaths during the vaccinal period, net of environmental temperature, seasonality, and annual trend. The corresponding regional excess all-cause deaths expected each day *d* of the vaccinal period and stratified for age category (${ED}_{rad}^{e}$) were calculated by subtracting the number of expected deaths (estimated from the SARIMAX model) from the number of observed deaths during the same period. Rates were standardized for the common age distribution of the population aged 20 years or older from Italy as reference (direct standardization) and expressed as excess deaths per 10,000 person-days. Further details of the time series model used in the current application are presented in Appendix A.

### 2.4. Prevented Fraction: Rationale and Estimate

Because vaccination campaigns against COVID-19 (the exposure) are believed to protect against excess all-cause deaths (the outcome), the prevented fraction for a population is the proportion of the hypothetical total load of the outcome in the population that has been prevented by the exposure [41], (i.e., the ratio between excess deaths prevented by vaccination and excess deaths expected to occur under the counterfactual of no vaccination). As neither the numerator nor denominator are directly known, an equivalent formulation can be derived from the algebraic relation of the prevented fraction with the prevalence of vaccinated individuals and vaccine effectiveness (VE) [42]. 

The first quantity, i.e., the cumulative prevalence of individuals belonging to a given age category (with *a* = 1, 2, or 3 for persons 20–49, 50–79, and 80-years or older, respectively, at baseline) reached by *v* = 1, 2, or 3 vaccine doses, everyday *d* of the vaccinal period (*PV_avd_*), was directly obtained from the available data [19]. 

The second quantity, i.e., VE in preventing death, is expected to vary over time because it (i) is reported to be different for the first, second, and third doses of the vaccine [13,14]; (ii) decreases over time after vaccine administration [7,8]; and (iii) is reduced by the replacement of the original virus with its variants [43,44]. In addition, VE is expected to vary according with age (better protective action expected at younger ages [45]). Therefore, VE was modelled according to the number of days (indexed as j) elapsed since administration of the *v*th vaccine dose and the prevalent variant circulating during the day *d* (*VE_jvd_*). These quantities were assembled for calculating the cumulative prevented fraction (PF) from 1 February 2021 (14 days after the first Italian citizen received the first COVID-19 vaccine dose; *d* = 0), until a given day *d*, to 31 October 2022, separately for each region (*r*) and age category. Further details on the sources and parameters used for PV and VE estimates, and the methods used for PF modelling, are reported in Appendix A.

### 2.5. Estimating the Rate of All-Cause Deaths Avoided

The cumulative number of excess deaths prevented by the vaccination campaign (*EDa*) implemented in each region from 27 December 2020 (*d* = 0), until a given day *d*, can be calculated as [46]:$${ED}_{r}^{a}=\sum_{a=1}^{3}\sum_{d=1}^{D}{ED}_{rad}^{e}\cdot \left(\frac{1}{1-{PF}_{rad}}-1\right)$$
with *a* = 1, 2, or 3 for persons 20–49, 50–79, and 80-years or older, respectively, at baseline, *r* = 1, 2, and 3 for citizens from Lombardy, Marche, and Sicily, respectively, and PF representing the cumulative prevented fraction until a given day *d*.

The rate of avoided deaths, standardized for the common age distribution of the population aged 20 years or older from Italy as reference (direct standardization), was expressed as the number of avoided deaths per 10,000 person-years and compared among regions.

The robustness of the standardized rate of avoided deaths was verified by changing some assumptions used for calculating the PF, as specified in Appendix A.

## 3. Results

### 3.1. Trends in SARS-CoV-2 and Excess Deaths

Throughout the study period, among citizens from Lombardy, Marche, and Sicily (i) 3,310,603, 614,375, and 1,422,041 positive swabs were detected (Appendix A), respectively, corresponding to positivity rates of 0.22, 0.27, and 0.20 per 10,000 person-years; (ii) 18,049, 5429, and 16,557 excess deaths were estimated, respectively, corresponding to excess death rates of 12, 24, and 23 per 10,000 person-years. Figure 1 compares the regional daily trends in positivity rate for nasopharyngeal swabs and expected excess deaths. Regarding swab positivity (upper box), low rates were observed until December 2021, followed by an initial sharp peak between December 2021 and January 2022 (reaching positivity values of 61, 62, and 36 per 10,000 person-days in Lombardy, Marche, and Sicily, respectively) and three other peaks of smaller intensity in April, July, and October. Regarding mortality (bottom box), lower rates were observed in Lombardy, with the highest value of 1200 excess deaths per 10,000 person-days reached a few days after the peak swab positivity of December 2021. The corresponding rates observed for that period in Marche and Sicily were 1600 and 2400 excess deaths per 10,000 person-days, respectively. Higher volatility in rates was observed in Marche and Sicily compared to Lombardy.

### 3.2. Trends in the Speed of Vaccination Campaigns and Prevented Fraction

The number of vaccination doses inoculated in the three regions over time is reported in Appendix A. Figure 2 shows a progressive reduction over time in COVID-19 vaccination campaign intensity from Lombardy (with an average of 1.82 doses received per citizen by October 2022) to Marche (1.67 doses) and Sicily (1.56 doses). Accordingly, a progressive reduction in the all-cause deaths PF from Lombardy to Marche and Sicily (corresponding PFs of 64.8%, 60.4%, and 58.0%, respectively, by October 2022) was also estimated (Figure 3).

### 3.3. Trends in the Rate of All-Cause Deaths Avoided

In the three regions combined, 104,513 all-cause excess deaths would have occurred without COVID-19 vaccination (42 per 10,000 person-years), while 40,036 all-cause excess deaths were observed during the investigated period (16 per 10,000 person-years), resulting in a gain of 64,477 saved deaths (26 per 10,000 person-years), which corresponds to an overall PF of 61.7%. The pattern of observed and avoided deaths was apparently similar in the three investigated regions (Figure 4). However, since Lombardy is characterized by a lower rate of all-cause excess mortality than Marche and Sicily (12, 24, and 23 per 10,000 person-years, respectively), despite a higher prevented fraction due to a more intensive vaccination campaign, the resulting rate of avoided deaths is lower (22, 36, and 32 per 10,000 person-years, respectively).

The patterns of observed and avoided deaths did not substantially change by modifying different VE values (Appendix A).

## 4. Discussion

Because of the between-country heterogeneity and variation over time in the application of standards for the certification of COVID-19 as the underlying cause of death [47], the country-level and worldwide assessments of pandemic impact by comparing COVID-19 mortality must be considered biased [28]. In addition, such measurements systematically underestimate the pandemic burden, since pandemics affect mortality from causes not directly due to infection, so justifying the observation that excess deaths overcame deaths attributed to COVID-19 from two- to three-fold in the period 2020–2021 in the global and European WHO regions [28]. Comparing the benefits of preventive interventions against the pandemic through the reduction of COVID-19 mortality has been consistently misleading.

During the vaccinal period from December 2020 to October 2022, in the three Italian regions included in the study, 104,513 (42 per 10,000 person-years) all-cause excess deaths would have occurred without COVID-19 vaccination, and 40,036 (16 per 10,000 person-years) all-cause deaths in excess of those expected according to mortality in the pre-COVID-19 period occurred, the balance being 64,477 (26 per 10,000 person-years) saved all-cause deaths from the effect of COVID-19 vaccination (i.e., about two-thirds of the expected deaths were averted). Considering (i) an all-cause excess mortality rate of 12, 24 and 23 per 10,000 person-years, and (ii) a prevented fraction of all-cause deaths of 65%, 60% and 58% for Lombardy, Marche and Sicily, respectively, the corresponding rates of averted deaths obtained for each region were 22, 36 and 32 per 10,000 person-years.

By assuming that the three investigated regions are representative of the entire Italian country, by reproportioning the absolute values obtained in the three regions on the number of citizens of the Italian population (about 48 million beneficiaries of the NHS aged 20 years or older), it can be easily estimated that about 387,000 excess deaths would have occurred in Italy without COVID-19 vaccination, while 239,000 excess deaths were saved by the effects of the vaccination campaign. Meslé et al. reported that between December 2020 and November 2021, COVID-19 vaccinations prevented 469,186 deaths of older adults (51.5% of expected deaths) in 33 European countries, with PFs ranging from 5.6% in Ukraine to as much as 92.9% in Iceland, depending on the speed with which the countries achieved high vaccination coverage [48]. We add to the available evidence that (i) the high PF obtained in Italy concerns all adult citizens, not only the elderly and (ii) deaths for all-cause, not only those for COVID-19, are saved by COVID-19 vaccination. This implies that, although our PF fraction of all-cause deaths was of the same magnitude as that reported in the literature, the impact of COVID-19 vaccination campaign in avoiding deaths is expected to be much higher than previously thought. It should be noticed, however, that Italy enforced the strongest COVID-19 vaccination policy as compared to other European countries, making vaccination mandatory for all working activities entailing social interactions [49]. In addition, our findings extend the estimate of the impact of COVID-19 vaccinations to a longer period than so far considered. Aside from the reassuring result of an impact that substantially maintains that reported for the first year after starting the vaccination campaign, our study presents general methods for calculating the prevented fraction as a vaccine campaign progresses (e.g., decreasing vaccine protective action, promoting boosters and updating immunization, and emerging new variants). These methods can be reproduced also to other settings, as well as to future pandemics. Compared to the other regions, Sicily had the lowest swab positivity rate (0.20 positives per 10,000 person-years, versus values of 0.22 and 0.27 in Lombardy and Marche, respectively) as opposed to one of the highest excess mortality rates (23 deaths per 10,000 person-years; similar to Marche, but double that of Lombardy, with rates of 24 and 12 deaths per 10,000 person-years, respectively). We suspect that the positivity rate reflects the testing availability and the contact tracing ability, rather than the spread of the epidemic. Because we tried to isolate time-varying determinants of excess mortality, SARS-CoV-2-related factors are expected to explain the observed excess with our approach. In these conditions, the lower the excess mortality, the better the control capacity of the SARS-CoV-2 epidemic, the timely treatment of citizens who experience COVID-19 symptoms, and/or the care keeping of no-COVID-19 patients during the acute phases of the pandemic. It follows that, since Lombardy proceeded with a more intensive COVID-19 vaccination campaign, a higher prevented fraction of all-cause deaths was achieved. Nevertheless, due to a lower excess mortality rate observed in Lombardy compared to Marche and Sicily, a lower rate of averted deaths was obtained. In other words, the more intensive COVID-19 vaccination campaign observed in Lombardy resulted in a lower rate of all-cause avoided deaths. This may have occurred because it took place in a more favourable epidemiological setting characterized by a healthcare system in which a better control capacity of the SARS-CoV-2 epidemic resulted in a lower excess mortality. Other than vaccines, several studies showed a strong association between the impact of the speed at which non-pharmaceutical interventions (NPI) (including school closure, public gathering bans, and isolation and quarantine) are implemented during a pandemic and the mitigation of pandemic consequences, such as the reduction of disease transmissibility and mortality rates [50]. Moreover, prevention and mitigation measures, including new diagnosis and surveillance technologies, pharmaceutical measures for post-exposure prophylaxis, and NPI, should be developed in order to mitigate the impact of future pandemics induced by the viruses from the same family of SARS-CoV [51,52,53].

Our study also showed a heterogenous vaccination coverage between regions. This may be explained by different reasons, among which the availability of vaccine doses, the efficiency in the delivery process, adequate logistics, vaccine hesitancy among the population [54], and a well-structured communication plan which may foster vaccination uptake and reduce vaccine hesitancy [55].

Despite its strengths, this study has important limitations. The non-pandemic counterfactual trend in mortality is derived using historical data and is sensitive to the assumptions made in the forecast. The implicit assumption of the SARIMAX models we used is that seasonality, non-stationarity, and autocorrelation in mortality rates observed in the past persist into the pandemic period, which may be untrue. In addition, the nonlinear patterns concealed in a time series are not captured by the ARIMA model. Another limitation relates to lack of accounting for additional variables that may impact the all-cause deaths (such as seasonal influenza) and to the use of proxies as surrogates for the true exposure to the determinants of mortality. For example, the average regional daily temperature we used as a covariate in the SARIMAX models likely does not capture the heat waves (and/or extremes of cold) affecting specific areas. Yet, certain approximations for deriving the PFs, for example the changes over time of VE, impact the accuracy of the empirical estimates. Although the robustness of our estimates was empirically verified by several sensitivity analyses, we cannot exclude that some sources of uncertainty remain. Finally, the study included three regions with high variability with regard to the speed of vaccination with different doses, the intensity of SARS-CoV-2 spread and its variants, and other environmental factors (such as daily air temperature), thus representative of different setting and geographical areas. However, we could not exclude that the impact of vaccination on all-cause excess mortality could have been different in other Italian regions.

## 5. Conclusions

In conclusion, we estimate that since the start of COVID-19 vaccination, the lives of almost 64,500 citizens have been saved through immunization in the three Italian regions considered in this study. Large between-region differences in the impact of vaccination on all-cause mortality was observed in our study. Early and full implementation of adult vaccination was associated with the largest reduction in expected all-cause deaths. Moreover, our results highlight the fact that, further actions such as contact tracing ability, timeliness of care, keeping of no-COVID-19 patient care, and, currently, secondary prevention strategy focusing on high-risk patients, are important factors to further reduce all-cause mortality.

## Figures and Tables

**Figure 1 vaccines-12-00413-f001:**
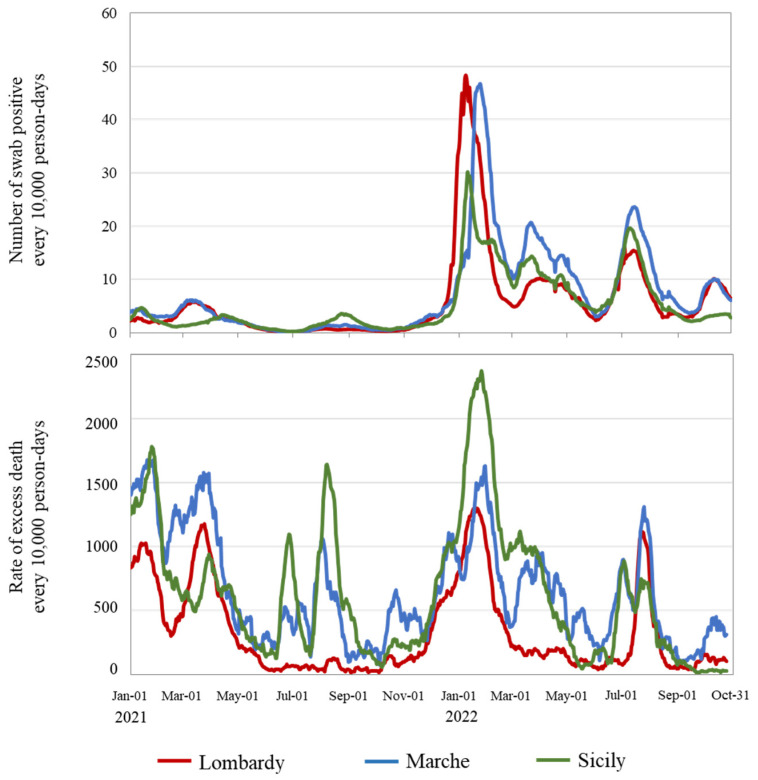
Daily time series of observed positivity rates for nasopharyngeal swabs (**upper box**) and expected excess death rates (**bottom box**) from 1 January 2021 to 31 October 2022, in the Italian regions of Lombardy, Marche, and Sicily. Footnote. Official data on citizens who tested positive on nasopharyngeal swabs for SARS-CoV-2 and all-cause deaths, provided by the Italian Health Ministry and the Italian National Institute of Statistics, respectively, were used. A simple moving average with 7-days width was used to attenuate the noise of positivity rates. A seasonal autoregressive integrated moving average with exogenous factors (SARIMAX) model was used to estimate the expected excess deaths, net of environmental temperature, seasonality, and annual trend. The corresponding rate, standardized for the common age distribution of the population aged 20 years or older from Lombardy as reference (direct standardization), was expressed as excess deaths per 10,000 person-days.

**Figure 2 vaccines-12-00413-f002:**
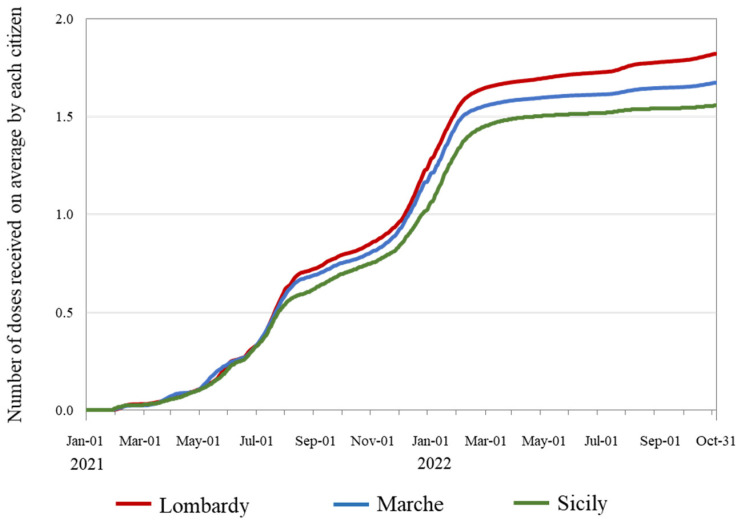
Daily time series of the average number of vaccine doses received by each citizen of the Italian regions of Lombardy, Marche, and Sicily from 1 January 2021 to 31 October 2022. Footnote. Official data on citizens who received one, two, or three COVID-19 vaccine doses, provided by the Italian Health Ministry, were used.

**Figure 3 vaccines-12-00413-f003:**
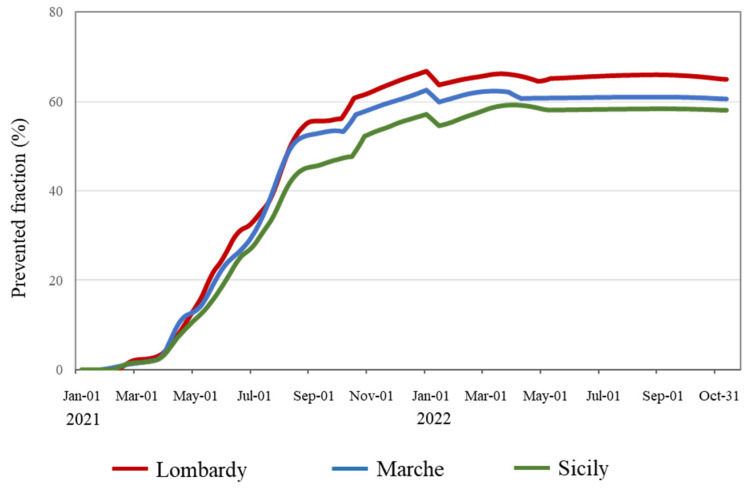
Daily time series of the prevented fraction from 1 January 2021 to 31 October 2022, in the Italian regions of Lombardy, Marche, and Sicily. Footnote. The prevented fraction was separately calculated within each region as the weighted mean of age-specific prevented fractions.

**Figure 4 vaccines-12-00413-f004:**
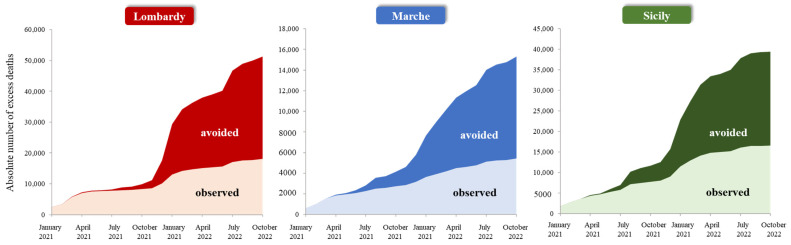
Daily time series of the cumulative number of excess deaths (i) observed (net of seasonality and temperature); (ii) avoided; and (iii) would have occurred without vaccination (obtained by considering the first and second quantities together) in the Italian regions of Lombardy, Marche, and Sicily.

## Data Availability

The datasets analyzed during the current study are publicly available. Mortality: https://www.istat.it/en/archive/deaths (accessed on 20 February 2023); COVID-19 vaccination campaign: https://github.com/italia/covid19-opendata-vaccini/tree/master/dati (accessed on 20 February 2023); Environmental temperature: https://scia.isprambiente.it/servertsdailyutm/serietemporalidaily400.php (accessed on 20 February 2023).

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
