# Peer review of "Estimating All-Cause Deaths Averted in the First Two Years of the COVID-19 Vaccination Campaign in Italy"

_vaccines, 2024, doi:10.3390/vaccines12040413_

Round 1

Reviewer 1 Report

Comments and Suggestions for Authors

I was invited to revise the paper entitled "Estimating all-cause deaths averted in the first two years of the COVID-19 vaccination campaign in Italy". It was a cohort study aimed to evaluate the avoided mortality due to cvoid-19 vaccination in three different Italian Region.

I want to congratulate with Authors for the well presented paper reporting crucial results for public health perspective.

Methodology was robust and clearly presented. Results are clear and easy to read.

Observations:

- Number of words seems to be poor. Authors should improve word count accordingly to journal indications;

- Introduction section should be improved, reporting how vaccination campign was developed in Italy during pandemic;

- In introduction section, Authors should also consider to evaluate the efficacy and effectiveness of Covid-19 vaccination in the Italian setting. Several paper were published covering all pandemic periods;

- Did Authors considered to perform a subanalysis by circulating variant of concern? for example comparing the omicron variant period to alpha variant period;

- In discussion section, Authors should discuss the possible bias derived  from the analysis of only three Region compaer to the rest of country. In addition, Authors should discuss the difference in vaccination coverages among Regions during the study period.

Reviewer 2 Report

Comments and Suggestions for Authors

The paper “Estimating all-cause deaths averted in the first two years of the 2 COVID-19 vaccination campaign in Italy” reports an estimation of deaths averted through vaccination in three Italian regions considering the prevented fraction. The approach is interesting even though uncertainty around data input can be relevant. This should be taken in due consideration by the Authors as they tend to overstate the importance of their results. Other major concerns are as follows:

Introduction

1.      I agree with the Authors with the ease and rationale of using all deaths in addressing the impact of COVID-19 vaccination campaign. Nevertheless, because of this, I do not understand why in the objective they talked about averted all-cause deaths directly and indirectly due to the SARS-CoV-2 virus. All deaths are not necessarily linked to SARS-CoV-2.

2.      Another point that I would ask the Authors to address is about the added value of their paper as the effectiveness of vaccination campaign has been proved by the Italian National Institute for Health throughout the vaccination campaign through the analysis of available data flows.

Methods

3.      Why did the Authors refer to population 20 years old or older? Furthermore, why did they stratify then the population in only three age groups?

4.      I have several doubts in respect to the calculation of the VE. Authors relied on ecological data, therefore the calculation of VE in respect to time elapsed from vaccination cannot be calculated as there is not any follow up information.

5.      It is confusing that the Authors used different denominators (100 person years, 10,000 person years, 10,000 person days, 1,000,000 person days for the calculation of rates). I would suggest having a unique denominator.  

Results

6.      The PF allows calculating deaths prevented by vaccination. This means that PF is not a result but a data input for the projection. Therefore, I would not expect to see it reported as a results in the sentence “resulting in a gain of 64,477 saved deaths (26 per 10,000 person-years), which 203 corresponds to an overall PF of 61.7%.”

Discussion

7.      In discussion Authors report data already provided in results. I would suggest shifting the attention at least on absolute numbers. This could make also more understandable the projection at national level that, anyway, should be supported by the explanation of methods behind it.

8.      Authors reported that the lower rate of all-causes avoided deaths in Lombardy was due the more favorable epidemiological setting that anyway was not due to vaccination campaign as said by the Authors.

Conclusion

9.      The conclusion is not supported by data as it refers to the projection to the national level that is not the objective of the paper.

Reviewer 3 Report

Comments and Suggestions for Authors

 Specific comments

·         Across the entire manuscript change “all-causes mortality” to “all-cause mortality

·         Lines 46-53: Here the authors could mention also the issue of reinfection during Omicron and protection provided by hybrid immunity with 1 or 2 doses against (symptomatic or asymptomatic) SARS-CoV-2 infection in observational studies (useful citation PMID: 37515237)

·         Line 130-142: It is unclear what measure of VE was eventually used to estimate the excess deaths prevented by vaccination.

·         Lines 133-136:In addition, VE is expected to vary according with age (better protective action expected at younger ages. Therefore, VE was modelled according to the number of days (indexed as j) elapsed since administration of the vth vaccine dose and the prevalent variant circulating during the day”… time since immunization until what event? Moreover, it is unclear why the latter time lag would remove the effect of age on COVID-19 mortality

·         Lines 133-134: VE is expected to vary also by comorbidity rate, e.g. Charlson index

·         Line 147: the equation and respective terms estimating the cumulative number of excess deaths prevented by vaccination should be defined/described

·         Line 159:100 person-years”… maybe it was meant to be “100,000 person-years”?

·         It is unclear whether the authors were able to account for the time lag elapsed since immunization and infection (14 since following first vaccine dose vs 7+ days following 2+ vaccine doses)

·         Line 185-186: “Figure 2 shows a progressive reduction in COVID-19 vaccination campaign intensity 185 from Lombardy”.. change to “progressive reduction over time in COVID-19 vaccination …etc.)

·         It is striking that that the excess of mortality was higher in all regions - especially in Sicily -  during January-February 2022, the first peak of transmission of the Omicron wave, a SARS-CoV-2 variant featured by high communicability yet low pathogenicity. This points should be commented.

·         Line 200: “Overall, in the three study regions”… in the three regions combined?

·         Line 207-208: This is unclear. If Lombardy had the lowest rate of all-causes excess mortality combined with highest prevented fraction, then the resulting rate of deaths avoided by vaccination should be higher (rather than lower)….

·         The results are only presented as Figures. Tables are recommended to visualize findings more in details. For instance, a descriptive table displaying the monthly vaccine coverage (0,1,2,34) by region would be useful for the reader.

·         238-240: It is unclear whether 387,000 would be excess deaths occurring without COVID-19 vaccination in the entire Italy.

·         Lines 253-54:our study presents methods for calculating the prevented fraction as the campaign progresses” Maybe the paper has been circulating for a while, this sentence is outdated.

·         Lines 261: “and the ability to properly track positives”… does that mean “contact tracing”?

·         Lines 270-71: “the more intensive COVID-19 vaccination campaign observed in Lombardy paradoxically resulted in a lower rate of all-causes avoided deaths”  this sentence should be mitigated as follows “…probably resulted in a lower rate of all-causes avoided deaths”

·         Lines 279-281:  intra-nasal administration of hypertonic saline solutions could be mentioned as pharmaceutical interventions for pre or post-exposure prophylaxis (recommended citations: PMID: 36432693; PMID: 37691090), at least to prevent/control the spread of mild/asymptomatic infections

·         In discussion it is essential to mention that Italy enforced the strongest COVID-19 vaccination policy across the EU, making vaccination mandatory for all working activities entailing social interactions. In all other countries vaccination was recommended but not mandatory, not even for health care workers.

·         Lines 301-302: Conclusions seem quite outdated with the current epidemiological scenario. Maybe the paper has been circulating for a while. Recommending high vaccination coverage and contact tracing is inappropriate and not not cost-effective now, considering the current infectivity yet low pathogenicity of the virus. A secondary prevention strategy focusing on high risk patients is more sensible. 

·        

Reviewer 4 Report

Comments and Suggestions for Authors

Dear Authors,

Thanks for submitting this manuscript that describes the findings from an ecological study that used a SARIMA model to estimate the impact of a COVID-19 vaccination campaign on all-cause mortality in three regions in Italy. The subject is of interest to the public health and wider healthcare community as your study findings add to the growing evidence base on the impact of vaccination programmes on disease morbidity and mortality when deployed alongside other prevention and control measures. 

I have outlined a few questions below and some suggestions to clarify some sections of the paper.

Page 2, line 87: can you please indicate why this study population was selected? Did you consider including children or were they not offered a COVID-19 vaccine in Italy?

Page 2, lines 93: "persons who tested positive by nasopharyngeal swabs for SARS-CoV-2" - do you mean both LFD and PCR positive swab tests, can you please specify as the diagnostic yield may have differed over time depending on the test being used.

Page 3, line 114: did the authors have the disaggregated data to allow a true incidence rate to be calculated or did they calculate incidence proportion?

Page 3, line 103 to 104: the authors describe the SARIMAX model in the main and supplementary papers and mention  the need to account for exogenous factors. This authors mostly focused on environmental factors and do not appear to have tried to account for concurrent provision/access to seasonal influenza vaccination and COVID-19 therapeutics among the study population. Was this considered and is there an explanation for why these factors were not included?

Page 3 and page 4: the authors have chosen to use different multipliers for the rate calculations, is there a reason for this beyond managing small numbers (Example: 1,000,000 person years for excess deaths, 10,000 person years for avoided deaths and 100 per person years for positivity rate.) ?

Page 4, lines 164-165: can the authors consistently use the same multiplier when reporting the same indicator (example: positivity rate is reported as 100 per person years in page 3 and then 10,000 per person years in page 4). 

Comments on the Quality of English Language

Satisfactory

Round 2

Reviewer 1 Report

Comments and Suggestions for Authors

Authors properly addressed all my previous comments.

Author Response

Thank you for your kind reply

Reviewer 2 Report

Comments and Suggestions for Authors

I thans the Authors for the provided clarifications and improvements. 

Author Response

Thank you for your kind reply

Reviewer 3 Report

Comments and Suggestions for Authors

The authors made an effort to clarify some points and addressed others, overall improving the manuscript. However, there are still a few comments which have been not been adequately/fully addressed . 

·        page 8, lines 278-281: This is not enough accurate, as words have a meaning. Italy not only enforced a “strong” , rather the “strongest” vaccination campaign across the entire Europe, as no other country enforced COVID-19 vaccination in order to be able to go to work. No other country suspended the registration of unvaccinated HCWs from the respective professional councils.

·        Some descriptive tables are recommended IN ADDITION to Figures

·        page 9, lines 343-353: this sentence is still not enough marked in describing a changing approach/scenario from primary prevention (until early 2022) to secondary prevention (thereafter)

·        Lines 299-302: “This likely occurred because it took place in a more favourable epidemiological setting characterized by a healthcare system in which a better control capacity of the SARS-CoV-2 epidemic resulted in a lower excess mortality”… this conclusion should be mitigated, e.g.  as follows: “This MAY HAVE OCCURRED because of a more favourable epidemiological setting and a better control capacity of the pandemic by the health care system, resulting in a lower excess mortality

The available information is a more intense vaccination campaign combined with lower mortality rate of Lombardy compared to other 2 regions and a slightly lower swab positivity rate in Sicily. However, a few lines above the authors used the expression “We SUSPECT the high morality rate etc.”

·        Reference 51 is a bit vague to back the concept of pharmaceutical interventions to prevent/control COVID-19, especially formulations targeting the nasal cavity and the upper airways more in general as pre/post exposure prophylaxis (again, recommended citations: PMID: 36432693; PMID: 37691090; PMID: 30705369).

Author Response

Specific comments:

  • page 8, lines 278-281: This is not enough accurate, as words have a meaning. Italy not only enforced a “strong” , rather the “strongest” vaccination campaign across the entire Europe, as no other country enforced COVID-19 vaccination in order to be able to go to work. No other country suspended the registration of unvaccinated HCWs from the respective professional councils.

Re: We modified the sentence as suggested by the reviewer.

  • Some descriptive tables are recommended IN ADDITION to Figures

Re: We added descriptive tables in the Supplementary Material.

  • page 9, lines 343-353: this sentence is still not enough marked in describing a changing approach/scenario from primary prevention (until early 2022) to secondary prevention (thereafter)

Re: We modified the sentence in order to better describe the changing scenario from primary to secondary prevention.

  • Lines 299-302: “This likely occurred because it took place in a more favourable epidemiological setting characterized by a healthcare system in which a better control capacity of the SARS-CoV-2 epidemic resulted in a lower excess mortality”… this conclusion should be mitigated, e.g. as follows: “This MAY HAVE OCCURRED because of a more favourable epidemiological setting and a better control capacity of the pandemic by the health care system, resulting in a lower excess mortality”

Re: We modified the sentence as suggested by the reviewer.

  • The available information is a more intense vaccination campaign combined with lower mortality rate of Lombardy compared to other 2 regions and a slightly lower swab positivity rate in Sicily. However, a few lines above the authors used the expression “We SUSPECT the high morality rate etc.”

Re: The term “suspect” is only reported in the manuscript with reference to “the positivity rate may reflect the testing availability and the contact tracing ability, rather than the spread of the epidemic”.

  • Reference 51 is a bit vague to back the concept of pharmaceutical interventions to prevent/control COVID-19, especially formulations targeting the nasal cavity and the upper airways more in general as pre/post exposure prophylaxis (again, recommended citations: PMID: 36432693; PMID: 37691090; PMID: 30705369)

Re: We replaced the reference 51 with those suggested by the reviewers.